# The Development and Implementation of Innovative Blind Source Separation Techniques for Real-Time Extraction and Analysis of Fetal and Maternal Electrocardiogram Signals

**DOI:** 10.3390/bioengineering11050512

**Published:** 2024-05-19

**Authors:** Mohcin Mekhfioui, Aziz Benahmed, Ahmed Chebak, Rachid Elgouri, Laamari Hlou

**Affiliations:** 1Green Tech Institute (GTI), Mohammed VI Polytechnic University, Benguerir 43150, Morocco; 2Faculty of Science, University Ibn Tofail, Kenitra 14000, Morocco; 3ERSC Team, Mohammadia Engineering School, Mohammed V University, Rabat 10106, Morocco; 4Laboratory of Electrical Engineering and Telecommunications Systems, ENSA, Ibn Tofail University, Kenitra 14000, Morocco

**Keywords:** ECG, blind source separation, IoT system, fetal ECG, real time implementation, heart rate, biomedical systems

## Abstract

This article presents an innovative approach to analyzing and extracting electrocardiogram (ECG) signals from the abdomen and thorax of pregnant women, with the primary goal of isolating fetal ECG (fECG) and maternal ECG (mECG) signals. To resolve the difficulties related to the low amplitude of the fECG, various noise sources during signal acquisition, and the overlapping of R waves, we developed a new method for extracting ECG signals using blind source separation techniques. This method is based on independent component analysis algorithms to detect and accurately extract fECG and mECG signals from abdomen and thorax data. To validate our approach, we carried out experiments using a real and reliable database for the evaluation of fECG extraction algorithms. Moreover, to demonstrate real-time applicability, we implemented our method in an embedded card linked to electronic modules that measure blood oxygen saturation (SpO2) and body temperature, as well as the transmission of data to a web server. This enables us to present all information related to the fetus and its mother in a mobile application to assist doctors in diagnosing the fetus’s condition. Our results demonstrate the effectiveness of our approach in isolating fECG and mECG signals under difficult conditions and also calculating different heart rates (fBPM and mBPM), which offers promising prospects for improving fetal monitoring and maternal healthcare during pregnancy.

## 1. Introduction

Birth defects are a leading cause of infant mortality, chronic disease, and disability in many countries. The 2010 World Health Assembly resolution encourages Member States to attach great importance to the prevention and health of children affected by these anomalies. Fetal death in the womb is an emotionally devastating event that affects parents and healthcare professionals. Newborn babies must be carefully monitored as congenital heart disease can manifest itself in the early stages of pregnancy. Rapid diagnosis is crucial because certain diseases, whether hereditary or not, can have serious repercussions. Approximately 6000 newborns are born each year in Morocco with cardiac anomalies or congenital heart disease [1], and prenatal pathology is generally diagnosed only after the onset of symptoms in the child, which may be too late. For this reason, fetal ECG analysis and monitoring have become more important than ever.

An electrocardiogram (ECG) is a technique for recording the electrical activities generated by the heart. Clinicians can assess the heart conditions of a patient through the use of an ECG and conduct a more thorough analysis or diagnosis. An ECG signal is generally measured from electrodes placed on the skin. In the case of a pregnant woman, acquiring fetal ECG becomes challenging because direct contact with the fetus is perilous [2,3]. In an electrocardiographic signal, the processes of myocardial contraction and relaxation are seen as a sequence of positive and negative deflections superimposed on a zero potential line (baseline) that corresponds to the absence of cardiac phenomena. Morphologically, adults and fetuses have fairly similar ECG patterns, but the relative amplitudes of the fetal complexes undergo considerable changes throughout gestation and even after birth. One big change concerns the T waves, which are rather weak for fetuses and newborns. An ECG not only provides information on heart rate but also on the position of the heart, the origin of the potentials, and the propagation of the depolarization wave. It is an essential clinical examination for exploring cardiac function and identifying any rhythm or conduction disorders.

Furthermore, ECG measurements obtained from the surface of a maternal abdomen contain various bioelectrical potentials, such as maternal cardiac activity, fetal cardiac activity, maternal muscle activity, fetal activity, noise, etc. These signal differences increase the difficulty of reconstructing a fetal ECG. Several studies have been conducted in this field based on different signal separation methods.

Blind source separation, specifically independent component analysis (ICA) [4,5], is the most widely published and used non-adaptive method for extracting fECG signals. This method assumes that the components are statistically independent and requires as many electrodes placed on the maternal abdomen as the number of uncorrelated signal sources. Therefore, when extracting fetal and maternal components from the abdominal signal, ICA requires a minimum of two electrodes. It is not necessary to use too many electrodes as each electrode carries its noise. During ICA preprocessing, centering is applied, making the vector a zero-mean variable, and whitening is performed, creating a new vector with uncorrelated components and unit variances [6].

Previously, only a few studies have been conducted on the real-time implementation of blind source separation (BSS) to extract the fetal electrocardiographic from the surface of a maternal abdomen. M. A. Hasan et al. [7] have modeled VHDL-based algorithms for FPGA implementation to efficiently extract the FECG signal from abdominal ECG using neural networks. C. Chareonsak and all [8] proposed and implemented a cost-effective FPGA hardware architecture to enable real-time blind source separation (BSS) for the separation of fetal ECG signals from maternal ECG interference using a modified Torkkola’s algorithm based on independent component analysis (ICA). E. Torti and all [9] present a hardware solution built on the Altera Stratix V FPGA for real-time separation and accurate detection of fetal ECG from maternal ECG. The proposed system uses blind source separation for fetal ECG extraction. Danilo Pani and all [10] worked on a block-by-block tracking algorithm that uses blind source separation (BSS) to digitally and uniquely extract a fetal ECG from non-invasive real-time recordings, effectively overcoming permutation ambiguity. The method is implemented in an OMAP L137 embedded processor for real-time applications. Bhavya Vasudeva et al. [11] presented an FPGA-based fetal heart rate monitoring system using an adaptive least mean square filter (LMS-AF) for fECG extraction. Raj, A. et al. [12] presented a new GWO-SA algorithm that combines gray wolf optimization with sequence analysis to improve non-invasive fetal ECG extraction from overlapping maternal signals. Dash, S.S. and all [13] presented a robust approach combining empirical mode decomposition (EMD), independent component analysis (ICA), and FIR filtering proposed for extracting fetal ECG (fECG) signals from the recordings of pregnant women. This technique effectively separates maternal ECG (mECG) and other noise sources, enabling the accurate extraction of fECG, validated on simulated and real-world data, demonstrating high performance when assessed via API evaluation. Sarafan, S. et al. [14] described a new algorithm using the Ensemble Kalman Filter (EnKF) to efficiently extract a fetal electrocardiogram (fECG) from a single-channel abdominal ECG (aECG), demonstrating superior performance to existing methods, with the results obtained from PhysioNet clinical data. Shi, X. et al. [15] introduced an unsupervised multilevel fetal ECG signal quality assessment method using features based on entropy, statistics, and ECG signal quality index, as well as an autoencoder-based feature. The results demonstrated a weighted average F1 score of 90% in the classification of high, medium, and low-quality fetal ECG signal segments, facilitating fetal heart rate estimation after the elimination of low-quality signals. Subha T.D et al. [16] presented a method for extracting fetal electrocardiograms (FECGs) from maternal abdominal signals using an adaptive least-mean-square (LMS) filter, LabVIEW, and Spartan 3 FPGA. Boudet, S. et al. [17] developed deep learning models to detect maternal heart rate (MHR) and false signals (FSs) on fetal heart rate (FHR) recordings, achieving good performance levels and integrated these models into an open-source MATLAB toolbox for morphological analysis of fetal heart rates. There are also algorithms based on ANNs designed to process the signal and extract useful information, as in the case of [18,19].

On the other hand, to date, none of these studies have addressed the challenge of separating sources in real time using the Arduino card, which is known for being a freely available, low-energy-consuming, and low-cost material. This work is also limited to extracting fECG signals and not processing this information to calculate heart rate, for example, or other parameters related to ECG signals to enhance fetal monitoring by doctors.

Our application aims to separate fetal cardiac activities (fECG), characterized by their low amplitude, from those of the mother (mECG) in real time. The goal is to extract specific components of the ECG signal for efficient fetal monitoring during pregnancy and childbirth. The separation will be performed using the method developed in the previous article [20] using blind source separation algorithms and the Arduino DUE board. Additionally, electronic modules will be added to measure blood oxygen saturation (SPO2) and body temperature, as well as to transmit the data to a web server. The results will be displayed on a small OLED display and a mobile application.

## 2. Materials and Methods

### 2.1. System Architecture

This study proposes a smart solution for monitoring pregnant women and their fetuses based on the ECG signals recovered from the thorax since this signal presents a good indicator of the fetus’s state of health while reducing the potential of associated risks. The developed solution is energy-efficient and includes several stages and systems (Figure 1):**Acquisition Block**: The acquisition system consists of two electrodes, one placed on the abdomen and the other on the thorax of a pregnant woman. The signals captured by these electrodes will be sent in real time to a separation and processing system;**Separation and Communication System**: This block consists of a high-performance on-board board for extracting the ECG signal from the fetus and its mother based on blind source separation algorithms, then calculating the heartbeat and extracting information related to each signal, such as the duration of each ECG wave. The card also measures blood oxygen saturation using a SpO2 sensor, after which these data are displayed on an OLED display and sent to an online database to facilitate fetal monitoring online or via a smartphone application;**Supervision System**: This is a smartphone application that serves as a secure platform, enabling not only parents but also the family doctor to remotely monitor and supervise the state of health of the fetus and its mother. Equipped with an alert system and full tracking of historical data, it provides a real-time overview of vital health parameters, promoting better care and peace of mind for families expecting a fetus.

### 2.2. ECG Fetal

Fetal ECG (fECG) is a non-invasive method of monitoring fetal heart activity. It is particularly useful during pregnancy when evaluating the state of health of the fetus and detecting any cardiac disorders. fECG can be measured by placing electrodes on the mother’s abdomen and chest/thorax (Figure 2) [21]. Fetal ECG is a recording that represents a method of monitoring the electrical activity of the fetal heart. It is a potential indicator of the state of health of the fetus and can change according to various internal and external events. These events have a significant influence on the interpretation of the fECG. That is why it is important to measure it accurately [22,23].

Several sources of noise and interference are added to the fetal ECG. These include fetal brain activity, the mother’s electromyograms (EMGs), respiratory activity, movement of the uterus, and disturbances (50 Hz) due to the mains. Furthermore, its variability depends on gestational age, the position of the electrodes, skin impedance, etc. Nevertheless, the main contamination is the ECG of the mother (mECG), whose amplitude is much greater than that of the fetus. The amplitude of the mECG changes during pregnancy, increasing during the first 25 weeks, experiencing a marked minimum around the 32nd week, and then increasing again. Consequently, the basic problem is extracting the fECG from the composite ECG, in which the fECG accounts for only 25% of the energy of the composite signal.

### 2.3. Composite ECG of a Pregnant Woman

When the electrocardiogram (ECG) of a pregnant woman is recorded on the abdomen, the obtained signal is a composite (Figure 3). It includes the mother’s ECG (mECG) to which the fetal ECG (fECG) is added, along with other signals resulting from uterine movements. The analysis of fetal ECGs has now become a routine medical procedure during pregnancy. A non-invasive technique for obtaining FECGs involves extracting it from the composite ECG of the mother, which is recorded at the abdomen. The challenge in this process lies in the fact that the fetal ECG is weak and is submerged within that of the mother [24,25].

### 2.4. About Blind Source Separation

Blind source separation (BSS) is a method of extracting the original signals from mixed signals without prior knowledge of the mixing process or the specific properties of the signals concerned. This approach applies to a wide range of data, including audio, visual, and biomedical signals [26].

The term blind denotes the fact that there is no a priori information about the sources or how they have been mixed. In the general case, three conditions are necessary when performing this technique:The sources are statistically independent;The number of sensors is higher or equal to the number of sources;A mixing matrix between the sources and the sensors.

An essential aspect of BSS is the identification of an effective measure for assessing the independence of separate signals. Various strategies have been developed to meet this challenge, including principal component analysis (PCA), independent component analysis (ICA), and non-negative matrix factorization (NMF). These techniques use distinct assumptions and computational approaches to develop a transformation that improves signal independence.

The principle of the BSS is depicted in Figure 4. In an ideal case, the principle is to find the matrix A of size Q × P, which provides the output vector:***y***(***k***) = ***W x***(***k***) = ***W H s***(***k***) ≈ ***s***(***k***)(1)
where:***x***(***k***): the vector containing the observations;***s***(***k***): the vector containing N signals emitted by N unknown sources;***y***(***k***): the vector of estimated sources;***H***: mixture matrix of size Q × P;***W***: the estimated Matrix *H* = *W*^−1^.

**Figure 4 bioengineering-11-00512-f004:**
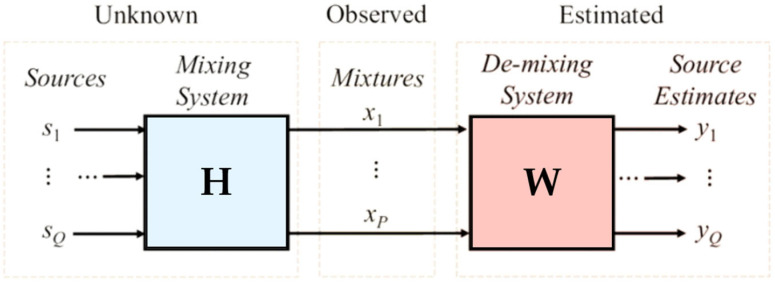
The figure shows the principle of the separation of sources [20].

The estimated sources are obtained from the vector *s*(*k*), and their related projections for the various sensors are determined from the estimated mixture array: H = W^−1^ [20,27].

## 3. Implementation Results

### 3.1. Database

The data used to validate this method were extracted from the FECGSYNDB database [28,29,30], a reliable source for testing fetal ECG extraction algorithms. The FECGSYNDB database contains ECG recordings from a pregnant woman using two electrodes: one placed on the abdomen and the other on the chest. The sampling duration is four seconds at a sampling frequency of 250 Hz, resulting in a signal of 1000 samples. The database utilized for the algorithm is presented in Figure 5. As seen in the samples, the data contain six beats of maternal ECG (mECG) and eleven beats of fetal ECG (fECG), where the mECG has a much higher amplitude than the fECG.

The signal of the fECG, if taken from the abdominal region, will exhibit very slight variations due to the relative distance between each electrode and the center of the fetal heart. This characteristic is not preserved when dealing with the mECG, as the signal will change significantly depending on the placement of electrodes on the abdomen.

By utilizing these characteristics, the mECG can be distorted through the combination of signals from different electrodes while preserving the fECG with minor changes. The use of a larger number of electrodes would pose a problem for real-life applications of this method. Thus, it is not necessary to use more than two electrodes for this procedure to facilitate ease of use and electrode placement. The combination of both signals will yield a signal with a high power ratio between the fECG and the mECG, simplifying the isolation process.

### 3.2. Hardware Implementation

In real-world applications, ECG signals are captured by electrodes placed on the abdomen and chest of the pregnant woman and then sent to the Arduino Due board [31] via the analog pins; alternatively, they can be stored on the board or sent from Matlab using the ‘*from workspace*’ block in the ‘*external mode*’ for simulation. A diagram of the separation, heart rate, and SpO2 monitoring is shown in Figure 6. The estimated signal (y_1_ and y_2_) and the original recording from the database of signals from the abdomen and maternal thorax (x_1_ and x_2_) are displayed in Figure 7. Signal y_1_ represents the extracted fetal ECG from signals x_1_ and x_2_ using the proposed algorithm, while signal y_2_ represents the ECG signal of the pregnant woman. It can be observed that the proposed method can extract a clean fetal ECG signal in real time without the loss of R-waves.

On the OLED display connected to the electronic board, it can be seen that the separation algorithm is SYM-WHITE [32,33] with a performance index [20] of 0.0012. The results of actual implementation demonstrate superior performance in extracting fECG and mECG signals without the need for a computer or separation software.

After separation, essential parameters such as the maternal heart rate (mBPM), the fetal heart rate (fBPM), the blood oxygen saturation level (SpO2), and the body temperature (T) of the pregnant woman are calculated. In addition, a sub-algorithm analyzes the ECG signal to determine critical intervals, such as the time between the R wave and the T wave (QT interval) and the variation in time between each R-R heartbeat (RR). This complete set of data is not only displayed on an OLED screen for immediate observation but also transmitted to a web server for remote access and continuous monitoring. By providing this detailed information, the system facilitates proactive healthcare interventions and ensures the well-being of both mother and baby throughout pregnancy and childbirth. The information sent includes the following:Maternal heart rate (mBPM);Fetal heart rate (fBPM);Oxygen saturation level (SpO2);Pregnant woman’s body temperature (T);The time interval between the R wave and the T wave (QT Interval);Variation in time between each R-R heartbeat (RR).

Figure 8 shows the architecture of the system developed in our laboratory, with a wristband integrated with a SpO2 sensor and an OLED screen, which is elegantly housed in a 3D-printed watch case. This portable system is complemented by modules dedicated to data acquisition and transmission. The communication module facilitates connectivity to Wi-Fi networks, enabling the real-time transmission of the acquired data to a designated web server. Users can easily access these data via a dedicated mobile application, improving the monitoring and management of vital health parameters.

Figure 9 provides an illustration of both the data structure and the real-time data within our Firebase database. The ‘*RT_DATA*’ table is designed to present the information obtained from the Wi-Fi module on a real-time basis, refreshing itself every minute. Concurrently, the data gathered each minute is systematically archived in the ‘**_history*’ table. These stored data are subsequently utilized for display purposes in the ‘*History*’ section of our application.

In order to make our data more accessible and easy to manipulate and to enable real-time monitoring, we have created a mobile application that runs on both the iOS and Android platforms. This application was created using Flutter’s Dart language and consists of two main pages. The first page is dedicated to user authentication, requiring a login and password. The second page displays real-time data on three tabs. These tabs are equipped with buttons for sharing data or contacting a doctor. The first tab displays ECG information for pregnant women. The second tab is dedicated to displaying fetal ECG information. Finally, the third tab, entitled ‘*History*’, displays the history of heartbeats (‘BPM’) over time. Figure 10 shows the real-time monitoring of results on the mobile interface, which offers the user three options:The first option enables exporting and sharing all data related to the ECG signals and the history of fBPM and mBPM.The second option enables sharing a screenshot of the real-time results, including the graph, via various connectivity tools, such as email or social media.The third option provides information about the mother and her fetus and technical support details.

The mobile application has been developed to send notifications and alerts to parents and the family doctor in emergencies or scenarios predefined by the doctor.

Based on the results obtained, this real-time ECG signal separation and monitoring system guarantees safe, reliable, and efficient fetal monitoring, and the separation and calculation history relating to the ECG signal can be monitored.

## 4. Discussion

The initial results have demonstrated that our developed IoT-based system provides users with a simple and comprehensive real-time monitoring system for the fetus and its mother. As today’s families are widely equipped with smartphones, this facilitates the use of our system. Given the importance of fetal monitoring to take measures in case of problems during childbirth and pregnancy, our system has low power consumption, making it environmentally friendly. The system prototype was successfully created and tested on a real database. These data are uploaded to the card via the USB port and will be processed in the same way if the data are received by the analog pins of the Arduino Due card (ADC1 and ADC2). Subsequently, the ECG signals will be separated by a set of blind source separation algorithms. In our case, the SYM-WHITE algorithm showed good separation performance, with a performance index equal to 0.0012.

After obtaining the fECG and mECG signals, digital filtering was applied to both signals to help smooth the curve so that the P, Q, R, S, and T waves were visible. Once the wave of the QRS complex is detected, peak counting is started to calculate the BPM. When counting peaks, it was a bit difficult to determine which peaks were being monitored and counted by the code, as some peaks were too low compared to others for each ECG signal. To facilitate detection, an autonomous horizontal reference line that calculates every 10 s was added. The duration of the RR and QT intervals was then calculated using an algorithm we developed to analyze the ECG signal. After calculating all parameters on several databases, we found that for a pregnant woman, the RR interval is between 600 and 1200 ms, the QT interval is between 350 and 450 ms, and the BPM is between 60 and 100. Similarly, for a fetus, the RR interval is between 300 and 600 ms, the QT interval is between 250 and 340 ms, and the BPM is between 105 and 150. Therefore, the program can also check whether the intervals fall within the health parameters. The smartwatch displays all of the calculated data in real time, along with the performance index. These data are updated every minute in a cycle to present the data of both the fetus and its mother. Likewise, the application presents the data with all the results shown, be it for separation or the calculation of parameters. Our system has consistently demonstrated good performance in real-time fetal monitoring.

## 5. Conclusions

This paper investigates the optimal configuration of a fetal growth monitoring system, using an integrated electronic card linked to a web server for online data storage and presentation on a mobile application to help the family physician diagnose fetal conditions.

A mobile application has been developed to facilitate tracking and history management. The proposed system is being extensively tested and validated using electrodes, sensors, and other display and data transmission modules via Wi-Fi.

The results showed that the system could successfully extract the fetal ECG signal from mixed signals with high accuracy and quality. The implemented algorithm can also estimate fetal heart rate, detect cardiac arrhythmias, and adapt to changes in fetal position and orientation, which can affect the mixing process.

The development and implementation of this blind separation and tracking system represent an important contribution to the field of biomedical engineering and signal processing. It represents a novel and effective solution for non-invasive fetal ECG monitoring during pregnancy. It has the potential to improve prenatal diagnosis and the management of various fetal cardiac pathologies. It can also improve the bond and communication between mother and fetus by enabling future parents to listen to their baby’s heartbeat.

## Figures and Tables

**Figure 1 bioengineering-11-00512-f001:**
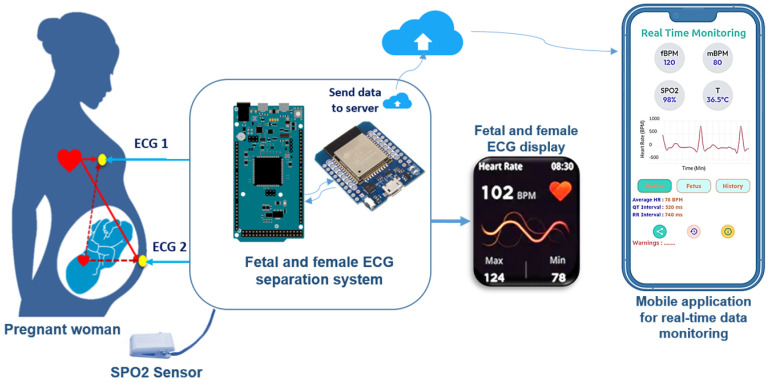
The figure shows the architecture of our intelligent system.

**Figure 2 bioengineering-11-00512-f002:**
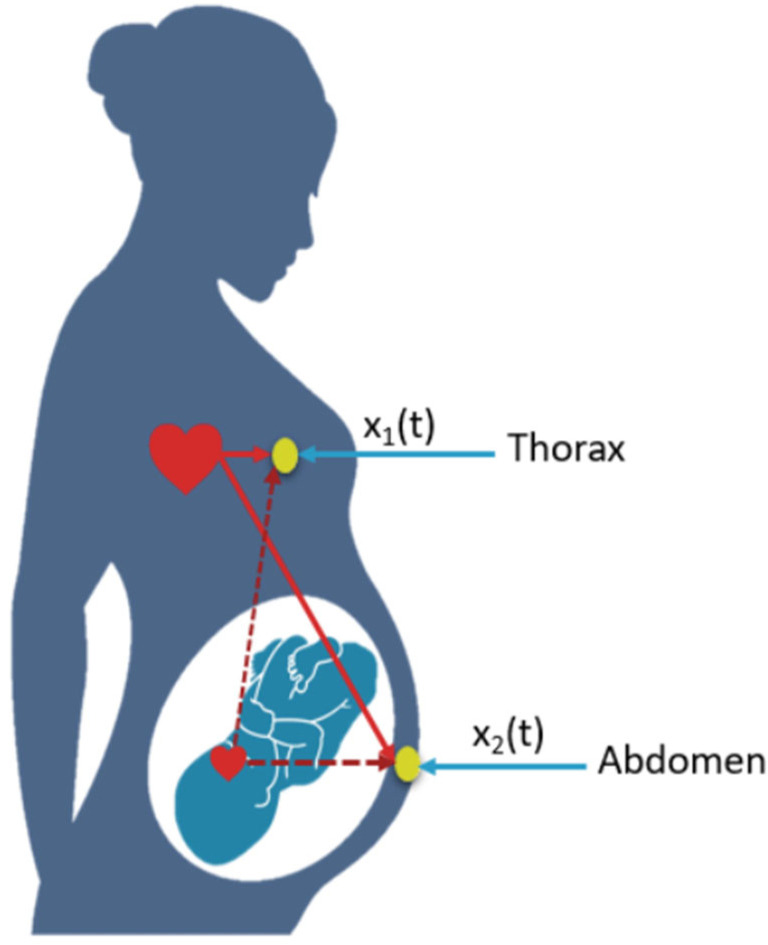
The figure shows the FECG recording method.

**Figure 3 bioengineering-11-00512-f003:**
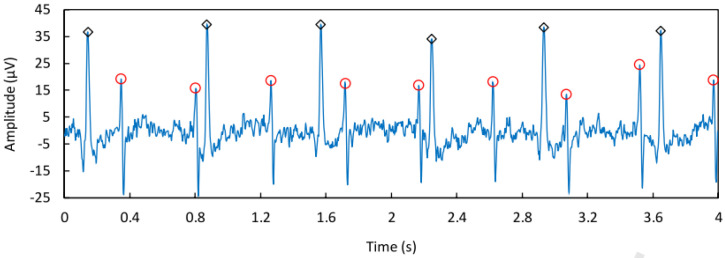
The figure shows an electrocardiogram of a pregnant woman. The red symbols indicate the QRS of the fetus; the black symbols indicate the QRS of the mother.

**Figure 5 bioengineering-11-00512-f005:**
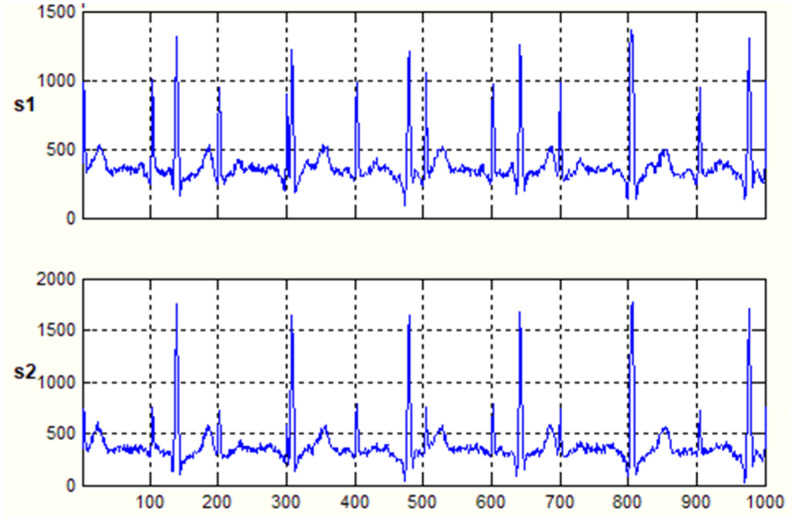
The figure shows an ECG of a pregnant woman (s1 from Thorax; s2 from Abdomen).

**Figure 6 bioengineering-11-00512-f006:**
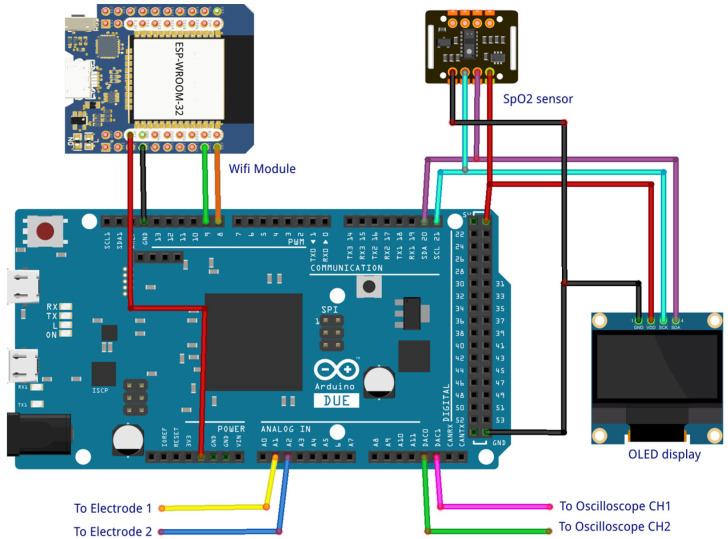
The figure shows the general connections of the complete setup.

**Figure 7 bioengineering-11-00512-f007:**
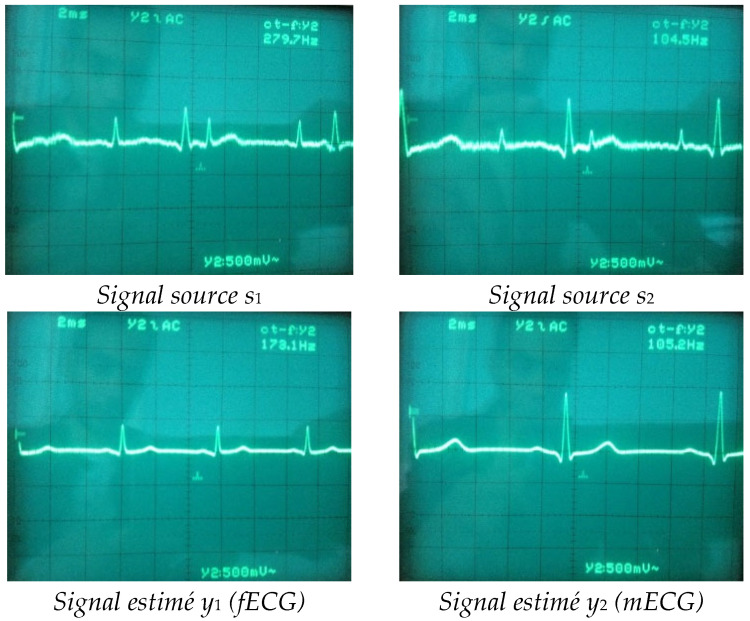
The figure shows the real-time extraction result.

**Figure 8 bioengineering-11-00512-f008:**
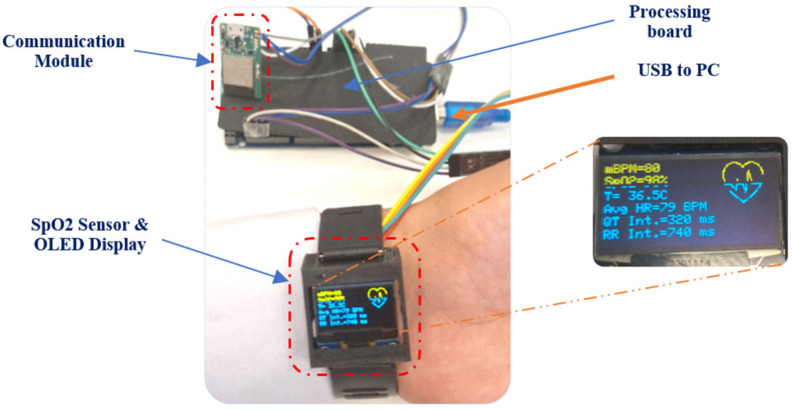
The figure shows the hardware setup for our system.

**Figure 9 bioengineering-11-00512-f009:**
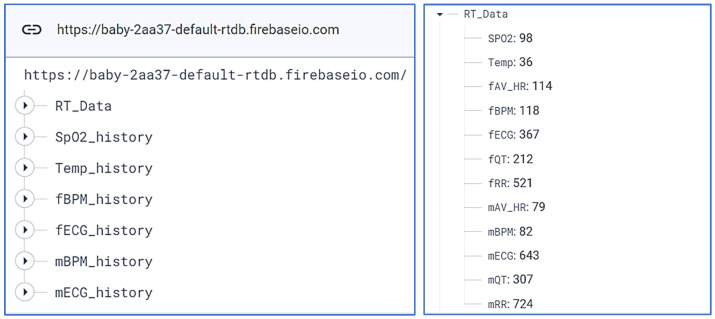
The figure shows our data in the firebase database.

**Figure 10 bioengineering-11-00512-f010:**
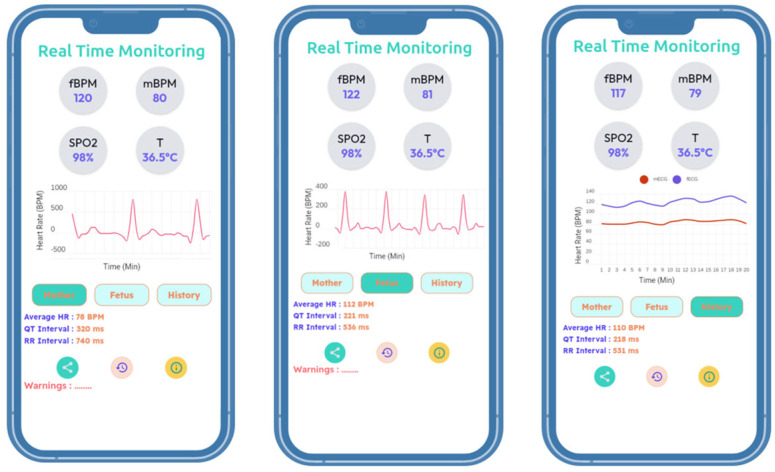
The figure shows the monitoring results on the mobile interface.

## Data Availability

The data presented in this study are available upon request from the corresponding author.

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
