# Peer review of "The Development and Implementation of Innovative Blind Source Separation Techniques for Real-Time Extraction and Analysis of Fetal and Maternal Electrocardiogram Signals"

_bioengineering, 2024, doi:10.3390/bioengineering11050512_

Round 1

Reviewer 1 Report

Comments and Suggestions for Authors

Authors have well organized the manuscript titled Development and Implementation of an Innovative Blind Source Separation Techniques for Real-Time Extraction and Analysis of Fetal and Maternal ECG Signals. Some modifications need to be done which are mentioned below.

1.     Mentioning performance in abstract may improvise your work

2.     Minor English language editing would improve the flow and the clarity of the text.

3.     Architecture of proposed method is shown in Figure 1 without mentioning the block. Including the block may help the readers for better understanding.

4.     Parameters in equation 1 need to discussed

5.     References are not mentioned properly as per the standard of the journal. Need to be modified.  Including recent citations may improvise the proposed model. Some of the recent works are mentioned below for your reference.

https://doi.org/10.1007/978-981-97-1335-6_21 

6.     In Figure 2, it is mentioned that black symbols indicate the QRS of the fetus and red symbols indicate the QRS of the mother. In Figure 4, it is mentioned that mECG has a much higher amplitude than the fECG which needs to clarified clearly.

Comments on the Quality of English Language

Moderate editing of English language required

Author Response

We sincerely thank you for taking the time to review our manuscript. Your insightful comments are enormously appreciated and will undoubtedly help to improve the quality and clarity of our work. Please find the detailed responses below and the corresponding revisions/corrections highlighted/in track changes in the re-submitted files.

Point-by-point response to Comments and Suggestions for Authors

Comments 1: Mentioning performance in abstract may improvise your work

Response 1: We agree, see lines 24-27

“Our results demonstrate the effectiveness of our approach in isolating fECG and mECG signals under difficult conditions, and also calculating different heart rates (fBPM and mBPM), which offers promising prospects for improving fetal monitoring and maternal healthcare during pregnancy.”

Comments 2: Minor English language editing would improve the flow and the clarity of the text.

Response 2: We have revised it, see the manuscript

Comments 3: The architecture of the proposed method is shown in Figure 1 without mentioning the block. Including the block may help the readers for better understanding.

Response 3: We have changed it, see the manuscript

Comments 4: Parameters in equation 1 need to discussed

Response 4: We have revised it,

?(?) = ? ?(?) = ? ? ?(?) ≈ ?(?)

Where

?(?): The vector containing the observations;

?(?): The vector containing N signals emitted by N unknown sources;

y(?): The vector of estimated sources;

H: Mixture matrix of size Q × P;

W: The estimated Matrix ? = ?−1.

Comments 5: References are not mentioned properly as per the standard of the journal. Need to be modified.  Including recent citations may improvise the proposed model. Some of the recent works are mentioned below for your reference.

Response 5: We have added it, see reference [10] in the manuscript

Comments 6: In Figure 2, it is mentioned that black symbols indicate the QRS of the fetus and red symbols indicate the QRS of the mother. In Figure 4, it is mentioned that mECG has a much higher amplitude than the fECG which needs to clarified clearly.

Response 6: Typing mistake, Figure 2 presents an Electrocardiogram of a pregnant woman. Red symbols indicate the QRS of the fetus; black symbols indicate the QRS of the mother.

Reviewer 2 Report

Comments and Suggestions for Authors

This article is lacking in some areas, especially:

- What amplifiers were used? What types of filters were implemented in the module?

- What was the ADC resolution? It is not stated in the article.

- Was the method tested on only data from the database or really captured by the system?

- The database is based on a model using 32 abdominal and 2 maternal ECG reference channels. How does it relate to lines 153-159 in Your article? It is not possible to obtain clear ECG signal without the use of, at least, a reference electrode for noise supression.

- How are the electrodes, arduino and smartwatch connected? Can You provide the schematics of the system? Figure 6 does not show any significant information. 

- Can You provide actual screencaps of the smartphone aplication or the web interface of the system? Figure 7 only shows the theoretical design of the GUI.

- Did You consider noise characteristics of the wireless interface of the system?

- Was the system tested on actual subjects? Or is it just a theoretical proposition?

- Please include references to other relevant signal analysis methods, ex. EEMD decomposition etc https://doi.org/10.3390/s22062176 https://doi.org/10.3390/s22103765

- How did You measure the effectiveness of the method? On what basis can You say that the method works or not? Results should be presented based on facts.

In my opinion the article should be significantly expanded as it provides very little actual detail. It seems that the research is not yet sufficiently advanced to be presented in the journal. 

Author Response

We sincerely thank you for taking the time to review our manuscript. Your insightful comments are enormously appreciated and will undoubtedly help to improve the quality and clarity of our work. Please find the detailed responses below and the corresponding revisions/corrections highlighted/in track changes in the re-submitted files.

Point-by-point response to Comments and Suggestions for Authors

Comments 1: What amplifiers were used? What types of filters were implemented in the module?

Response 1: The amplifier used is the AD8232 circuit, which performs amplification and filtering, and then uses blind source separation algorithms to separate the two ECG signals.

Comments 2: What was the ADC resolution? It is not stated in the article.

Response 2: ADC resolution is 12bits (ADCs on Atmel SAM3X8E ARM Cortex-M3 CPU board)

Comments 3: Was the method tested on only data from the database or really captured by the system?

Response 3: The proposed method is tested only on data from a real database extracted from the FECGSYNDB database, due to the difficulty of testing the system on a pregnant woman without official authorization.

Comments 4: The database is based on a model using 32 abdominal and 2 maternal ECG reference channels. How does it relate to lines 153-159 in Your article? It is not possible to obtain clear ECG signal without the use of, at least, a reference electrode for noise supression.

Response 4: Using blind source separation algorithms is based on the assumption that the number of sensors must be greater than equal to the number of sources, in our case the number of sources is two (mECG and fECG), so we can't use just one electrode.

Comments 5: How are the electrodes, arduino and smartwatch connected? Can You provide the schematics of the system? Figure 6 does not show any significant information.

Response 5: Our smartwatch is connected to the arduino board via I2C communication, and the electrodes will be connected via pins ADC1 and ADC2. In our tests, we sent the data for both electrodes from the computer (Matlab).

Comments 6: Can You provide actual screencaps of the smartphone aplication or the web interface of the system? Figure 7 only shows the theoretical design of the GUI.

Response 6: The screenshots in Figure 7 are captured from an Android/IOS emulator (flutter-flow) which reflects the real operation of the application that communicates with the Firebase server to display data in real-time.

Comments 7: Did You consider noise characteristics of the wireless interface of the system?

Response 7: Noise from the system's wireless interface is negligible, thanks to the internal structure of the ESP32 module to filters out noise and guarantee better information transformation.

Comments 8: Was the system tested on actual subjects? Or is it just a theoretical proposition?

Response 7: The system is tested on a real application and processing is done on a real electronic board as shown in Figure 6, except for the ECG data which are sent from Matlab and not with real electrodes given the difficulty of testing on a pregnant woman. 

Comments 9: Please include references to other relevant signal analysis methods, ex. EEMD decomposition etc https://doi.org/10.3390/s22062176 https://doi.org/10.3390/s22103765

Response 9: We have added it, see reference [10, 12, 13] in the manuscript

Comments 10: How did You measure the effectiveness of the method? On what basis can You say that the method works or not? Results should be presented based on facts.

Response 10: The efficiency of our method is evaluated by calculating the separability performance index (PI), which should be nearer 0. In our case, we found 0.0012, which means better separation, validated also by the separation images in Figure 5. 

Round 2

Reviewer 1 Report

Comments and Suggestions for Authors

Authors have answered the queries nicely and organized the paper properly. But, still some minor correction may improve the clarity of presentation for the potential readers. 

1. Fig.5 has two subfigures and needs proper caption for the same. 

2. Which ICA algorithm has been used by the method and why? 

3. How the mixing matrix is generated? 

4. How the performance of separation is evaluated? Provide some objective measures. 

5. Provide the detailed description of the sensors.

6. Clarity is missing about the subjects (gestation period) used for this study.

Author Response

We sincerely thank you for taking the time to review our manuscript. Your insightful comments are enormously appreciated and will undoubtedly help to improve the quality and clarity of our work. Please find the detailed responses below and the corresponding revisions/corrections highlighted/in track changes in the re-submitted files.

Point-by-point response to Comments and Suggestions for Authors

Comments 1: Fig.5 has two subfigures and needs proper caption for the same.

Response 1: We agree, see lines 226

“Figure 5. ECG of a pregnant woman (s1 from Thorax, s2 from Abdomen).”

Comments 2: Which ICA algorithm has been used by the method and why?

Response 2: The algorithm used for signal separation is the SYM-WHITE algorithm because it has a better performance index.

Line 248: “On the OLED display connected to the electronic board, it can be seen that the separation algorithm is SYM-WHITE [32,33] with a performance index [20] of 0.0012.”

Comments 3: How the mixing matrix is generated?

Response 3: The mixing matrix is unknown and we try to determine it using the SYM-WHITE algorithm.

Comments 4: How the performance of separation is evaluated? Provide some objective measures.

Response 4: Separation efficiency is assessed by calculating the separability performance index (PI), which should be close to 0. In our case, we found 0.0012, which means better separation, also validated by the separation images in Figure 7. 

Comments 5: Provide a detailed description of the sensors.

Response 5:

-        SpO2 Sensor: MAX30105 is a high-precision, low-power consumption, easy-to-use heart rate and blood oxygen saturation sensor. It is capable of measuring heart rates from 30 to 255 beats per minute and also measuring blood oxygen saturation levels from 70% to 100% with an accuracy of +/- 4%.

-        To capture the ECG signals, we used electrodes with an AD8232 amplifier, which is a low-power, low-noise instrument designed to measure biopotential signals. It has adjustable gain, a built-in low-pass filter, and ESD protection. It is a popular choice for health monitoring and medical diagnostic applications.

Comments 6: Clarity is missing about the subjects (gestation period) used for this study.

Response 6: Our objective is to create an intelligent, low-power embedded system with a smartphone application to separate fetal (fECG) and maternal (mECG) cardiac activity in real-time, and to extract specific components of the ECG signal for effective fetal monitoring during pregnancy and delivery.

Reviewer 2 Report

Comments and Suggestions for Authors

Dear Authors

Thank You for answering the questions from the first review. The additions in the article are, in my opinion, sufficient. Especially the explanation of using only previously recorded data from the database is important.

I have experience in designing and using biomedical signal amplifiers, and it is not an easy task. AD8232 is a simple module, which provides at most medium quality signal when using only two electrodes. I once performed experiments with it and I recommend avoiding the use of only two leads operation (figure 62 in the datasheet), as it is very sensitive to noise and generates poor output signal. Better focus on the circuit similar to the one on figure 66, as it is the most sensitive. Even if You are unable to use the device on actual patients, consider buying or building a circuit with a DAC, a differential output amplifier, synthetic skin and other elements to simulate the actual conditions.

Feel free to contact me if You need help. Actual complex bioelectric signal measurements is quite a task, and I might be able to provide advice if Your team wants to go in this direction. My contact data should be available when the article is published.

Best wishes

Author Response

Dear reviewer,

Thank you for your detailed comments and shared experience in the design and use of biomedical signal amplifiers. Your feedback is extremely valuable in improving our work.

Yes, exactly, we found it difficult to make an amplifier based on the AD8232, which is very sensitive to noise. We made a PCB board, and this improved the output signal a little while eliminating cable noise.

Your offer of support is also much appreciated. We'll make sure your contact details are available after the article is published.

Best regards.
